# Study of Ultrasonic Guided Wave Propagation in Bone Composite Structures for Revealing Osteoporosis Diagnostic Indicators

**DOI:** 10.3390/ma16186179

**Published:** 2023-09-12

**Authors:** Evgeny V. Glushkov, Natalia V. Glushkova, Olga A. Ermolenko, Alexey M. Tatarinov

**Affiliations:** 1Institute for Mathematics, Mechanics and Informatics, Kuban State University, Krasnodar 350040, Russia; nvg@math.kubsu.ru (N.V.G.); o.ermolenko.a@gmail.com (O.A.E.); 2Institute of Electronics and Computer Sciences, LV-1006 Riga, Latvia; alta2003@apollo.lv

**Keywords:** composite bone phantoms, guided wave modal excitability, restoration of effective elastic moduli, resonance diagnostics

## Abstract

Tubular bones are layered waveguide structures composed of soft tissue, cortical and porous bone tissue, and bone marrow. Ultrasound diagnostics of such biocomposites are based on the guided wave excitation and registration by piezoelectric transducers applied to the waveguide surface. Meanwhile, the upper sublayers shield the diseased interior, creating difficulties in extracting information about its weakening from the surface signals. To overcome these difficulties, we exploit the advantages of the Green’s matrix-based approach and adopt the methods and algorithms developed for the guided wave structural health monitoring of industrial composites. Based on the computer models implementing this approach and experimental measurements performed on bone phantoms, we analyze the feasibility of using different wave characteristics to detect hidden diagnostic signs of developing osteoporosis. It is shown that, despite the poor excitability of the most useful modes associated with the diseased inner layers, the use of the improved matrix pencil method combined with objective functions based on the Green’s matrix allows for effective monitoring of changes in the elastic moduli of the deeper sublayers. We also note the sensitivity and monotonic dependence of the resonance response frequencies on the degradation of elastic properties, making them a promising indicator for osteoporosis diagnostics.

## 1. Introduction

Among the developed quantitative ultrasound (QUS) approaches, the one based on guided waves relies on the waveguide properties of cortical long bones (e.g., see books [1,2] and reviews [3,4] for more details). This QUS method (often referred to as axial transmission) is applied to appendicular skeletal sites, such as the tibia and radius, and—to a lesser extent—the skull and phalanges. Ultrasonic-guided waves (GWs) are generated and registered by piezoelectric transducers applied to the soft tissue covering the cortical bone. The frequency response and dispersion characteristics of the traveling waves propagating from the source to sensors are mostly due to the specific geometry and elastic properties of the layered biocomposites: soft tissue–bone tissue–bone marrow. Since the ultrasound propagation parameters reflect the material properties, understanding the GW dependence on factors related to bone health helps to reveal hidden signs of osteoporosis. A recent and historical background on this topic is available in Ref. [5], while various aspects of QUS studies with both bio-mimicking samples and ex vivo samples can be found in Refs. [6,7,8,9,10,11,12,13,14,15,16,17,18].

Ultrasonometry has advantages over the widely used X-ray densitometry, such as the absence of ionizing radiation, compactness, and lower costs. There are also other methods for bone inspection based on various physical phenomena and principles, such as magnetic resonance imaging [19,20], pulse-echo measurements [21], and ultrasound back scattering [22]. They aim to assess bone porosity and thickness by measuring the free water content in the bone volume or the reflected and scattered waves.

The QUS of bone biocomposites represents a specific application of ultrasonic non-destructive testing (NDT) and guided wave structural health monitoring (SHM) technologies [23,24]; these technologies were developed to detect incipient defects and monitor changes (degradation) in the strength properties of laminate composite materials used in various industrial applications, e.g., carbon–fiber-reinforced polymers in aerospace units, pipelines, etc. One would expect QUS to operate by the same methods. But unlike traditional laminate composites fabricated from similar sublayers (prepregs), the sublayers of bone biocomposites have very different elastic properties. And since osteoporosis develops from inside the tubular bone [25,26], the degradation of inner sublayers is of prime concern.

The upper sublayers composed of soft and cortical bone tissue act as shields to the diseased interior, creating difficulties in extracting useful information about its weakening from signals $v_{i}\left(t\right)=v\left(x_{i},t\right)$ received on the surface (here, $v\left(x,t\right)={\dot{u}}_{z}\left(x,t\right)$ is the velocity of the normal component of the displacement vector $u=(u_{x},u_{y},u_{z})$ at a surface point x=(x,0,0)). Therefore, NDT and SHM methods need to be refined and improved to diagnose specific bone structures.

The identification of diagnostic features requires solutions to complex mathematical problems arising in the simulations of wave processes in multilayered samples (phantoms), mimicking waveguide properties of tubular bones. Phantoms are commonly used as substitutes for hard-to-find sets of real bone samples [8,10]. First, there are direct problems of GW propagation in the layered composite structures. Based on their solutions, it is possible to analyze the GW dependencies on the factors that indicate the presence or development of disease, e.g., the thickness and elastic properties of the cortical layer, the increase in porosity, the weakening of the inner sublayers, etc. Second, there are inverse problems in recovering the sublayer thicknesses and effective elastic parameters from data arrays $v_{ij}=v_{i}\left(t_{j}\right)$ of the digitized signals recorded at surface points $x_{i}$. The effective parameters are usually obtained by minimizing the discrepancy between the experimental and calculated wave characteristics, as the input material constants and sample geometry vary.

Nowadays, numerical simulations of elastodynamic behaviors of layered composites are conventionally performed using mesh or hybrid numeric–analytical methods, such as the finite element method (FEM) or semi-analytical finite elements (SAFEs). In our research, we develop a meshless semi-analytical approach based on the explicit integral and asymptotic representations in terms of the Green’s matrix of the composite structure considered. This not only reduces the computational costs but also provides direct insight into the wavefield structure by providing the amplitude and dispersion characteristics of each GW mode excited by the source. The developed algorithms of the Green’s matrix allow simulating the wave propagation in arbitrarily anisotropic [27,28], functionally graded [29], or fluid-filled porous [30] layered structures. Therefore, this approach has already proved its effectiveness in solving various NDT and SHM problems, such as crack detection [31] and time-reversed defect location [32], the restoration of effective elastic moduli of fiber-reinforced composite plates [33], nitride nanowire-based composite materials [34], trapped mode resonance identification [35,36], and others. The main objective of the current studies is to leverage the benefits of the Green’s matrix-based approach in identifying concealed diagnostic indicators of developing osteoporosis from recorded surface signals.

Since the experimental measurements provide arrays of recorded transient signals, a problem that emerges with the independent meaning arises: extracting the frequency characteristics of the excited GW modes, with a focus on their dispersion characteristics. Conventionally, the discrete Fourier transform over time *t* and distance *x* is applied to such arrays, yielding a so-called function H(α,f) that approximates the Fourier symbol V(α,f) of the signals’ field v(x,t) [37] (*f* is the frequency and α is the Fourier parameter for the transform over *x*, which is also a wavenumber for the waves propagating in the *x* direction). The local maxima of the *H*-function indicate dispersion curves $\alpha =\zeta_{n}\left(f\right)$, visualizing fragments of their trajectories in the frequency–wavenumber plane (f,α).

In recent years, an approach based on the application of the matrix pencil method (MPM) [38,39] to arrays $v_{ij}$ has gained popularity. It provides a set of desired points $\left(f_{p},\zeta_{p}\right)$ lying on the dispersion curves. Moreover, unlike the method of *H*-functions, the MPM yields complex wavenumbers $\zeta_{n}$. This allows quantifying not only the phase velocities cn=ω/Reζn of the excited GWs but also the logarithmic decrements δn=2πImζn/Reζn of their attenuation caused by the softening of bone tissue due to the increase in porosity. Hence, this characteristic can be considered as a potential diagnostic sign of osteoporosis development. The obtained pairs $\left(f_{p},\zeta_{p}\right)$ are used as experimental reference values in the objective functions of the inverse problem of determining the effective parameters. In QUS, similar approaches based on forming a response matrix from the acquired data arrays and finding its eigenvalues (wavenumbers) are being actively developed by the Laugier–Minonzio group [8,9,10,11] and other research teams (e.g., [17]). Apart from MPM, a singular value decomposition method is also used here to extract wavenumbers from the response matrix, but without their imaginary parts.

In our studies, to extract wavenumbers from the measurement data, a modified MPM [40] is used with additional filtering by the *H*-function [41]. The objective function of the inverse problem is expressed in terms of the Green’s matrix elements calculated at the points of GW dispersion curves selected from the recorded signals. Such an objective function, proposed in [34,42], reduces computational costs by 2–3 orders of magnitude compared to the conventional fitting of theoretically calculated and experimentally obtained dispersion curves.

The foundation of the present studies traces back to bone ultrasonometry research conducted in Riga starting in the 1980s [43,44]. In the 2000s, the research temporarily moved to Artann Laboratories, USA [7,45], and has since resumed in Riga [46], while the numerical analysis is carried out by the Krasnodar team [41,47,48].

The research objective of the present work is to search for and analyze wave characteristics, the change of which could indicate the development of osteoporosis. The effective material parameters obtained by solving the inverse problem directly specify the bone’s state. Therefore, their change can serve as a direct diagnostic indicator, while the change in dispersion curves can indirectly indicate the osteoporosis development. Another promising indirect diagnostic sign is a change in the pattern of resonance peaks in the frequency response. The diagnostic indicators revealed from using the developed computer model are discussed in Section 6, after describing the experimental technology (Section 2), mathematical model (Section 3), and data processing methods (Section 4 and Section 5). Section 3 also considers the possibilities of traditional dispersion curve indicators. It is shown that despite an appreciable variability in the theoretical dispersion curves, their real use is theoretically limited by the poor excitability of the ‘useful’ modes associated with the internal diseased sublayers.

## 2. Bone Phantoms and Experimental Measurements

Bone is a complicated biological composite that boasts a pronounced hierarchical structure, ranging from mineral–collagen to osteon levels. It is extremely difficult to artificially mimic the bone structure and the properties arising from it in close approximation, especially if the purpose is to provide its predictable grades. The primary aim of axial transmission measurements is to identify diagnostic indicators. These indicators become evident in the transformation of wave characteristics accompanying osteoporotic changes in bone properties. At the same time, the material properties themselves are not as important for wave studies as the relationships between body wave velocities and trends in their changes. By frequency tuning, it is possible to achieve the required wavelength-to-thickness ratios; therefore, the sample material does not necessarily have to provide the same body wave velocities as in the bone.

Experimental and theoretical studies are usually performed with artificial guides mimicking the wave properties of real bones, the so-called bone-mimicking phantoms [8,10]. To account for the cylindrical form of tubular bones, the phantoms are often fabricated in the form of layered pipes (e.g., [12]). However, the response of the bone’s wall to a localized surface loading is similar to that of an elastic plate, and comparative measurements show that laminate plates with properly chosen effective elastic properties can provide the same waveguide properties [8]. Such a replacement of tubular waveguides by plates is not a straightforward procedure, since the impact of bone curvature depends on the relations between the wall thickness, the outer diameter of the tubular bone, the driven wavelength, the probe characteristics, and many other factors. Nevertheless, it is widely used in research, for example, in the guided wave structural health monitoring of industrial pipelines [49].

Various aspects of sample selection (the effects of soft tissue coating, bone curvature, anisotropy, porosity, absorption, and so on) are thoroughly discussed in the studies by the Laugier–Minonzio group, e.g., [9,10]. For example, to account for porosity-induced anisotropy, a transversely isotropic composite of short glass fibers embedded in an epoxy matrix was used as bone-mimicking material [9]. On the other hand, the numerical analysis carried out in Ref. [48] indicates a minimal effect of accounting for such anisotropy on key wave characteristics, such as the patterns of dispersion curves of well-excitable GW modes in multilayered bone-mimicking samples with soft coating. Therefore, for the present studies, we used samples fabricated from isotropic poly(methyl methacrylate) (PMMA) plates (hereafter, PMMA will also be referred to as plexiglass).

Plexiglass has already been considered as an ultrasound reference material, especially considering its acoustic impedance closely aligns with that of bone [50], as well as its uniformity, and precision-shaping abilities [51,52]. In our studies, we trace the influence of the development of porosity from the inside to the periosteum. It is necessary to distinguish between the states of porous thin bone due to the peculiarities of the human constitution and different degrees of bone porosity caused by osteoporosis. To model these conditions with a high degree of reproducibility, a workable plexiglass is a good choice.

In the experiments, we used a set of phantoms made of 120 mm by 25 mm plexiglass plates with a thickness *h* from 2 to 6 mm, which is a typical thickness variation of the cortical layer in the metaphyses of large tubular human bones (Figure 1a,b). Soft organic tissue was modeled by a plastic layer of thickness $h_{soft}$ from 0 (no coating) to 5 mm, covering the plexiglass plate. Osteoporosis leads to a thinning of the cortical layer and an increase in intracortical porosity from the inner (endosteal) side [25,26]. To simulate this manifestation, 0.5 mm diameter holes were drilled from the bottom of the plates to mimic the effect of porosity by reducing about 20% of the material volume. The holes were drilled using a computer numerical control (CNC) machine with a programmable sequence of the holes. The drilling depth $h_{pore}$ varied from 0 (no pores) to the full plate thickness *h*, so that there were two sublayers of the thicknesses $h-h_{pore}$ and $h_{pore}$ with different effective densities and elastic moduli given in Table 1.

The measurements were carried out at the experimental setup (Figure 1, bottom) according to the measurement scheme shown in Figure 2. A contact piezo actuator (emitter), applied to the sample’s surface, produced a normal surface load $\sigma_{z}=q\left(x,t\right)$ that generated ultrasound GWs propagating along the sample. The signals $v_{i}\left(t\right)={\dot{u}}_{z}\left(x_{i},t\right)$ (velocity of the normal surface displacement component) were acquired at $N_{x}+1$ receiving points $x_{i}=x_{0}+i\Delta x$, $i=0,1,2,\ldots ,N_{x}$ (in most experiments, the distance from the emitter $x_{0}=50$ mm, the points’ spacing Δx=1 mm, and $N_{x}=23$). The acquired signals were recorded with a time increment Δt: $t_{j}=t_{0}+j\Delta t$, forming data arrays $v_{ij}=v_{i}\left(t_{j}\right)$; in the experiments, Δt= 0.03 μs.

Since the contact area was relatively small and, therefore, had little effect on the GW characteristics, the source was modeled by a point load: q(x,t)=δ(x)p(t). In the experiments and simulations, the driving impulse was taken either in the form of a modulated two-cycle sinusoidal pulse with a central frequency $f_{c}$:$$p\left(t\right)=\sin\left(2\pi f_{c}t\right)\sin\left(\pi f_{c}t/2\right),\;0\leq t\leq 2T,\;T=1/f_{c},$$
or as a sweep signal p(t)=sin(2πf(t)t) with a linear frequency decrease from f= 500 kHz to 50 kHz within 0.02 ms (Figure 3).

To illustrate the dynamic response of the phantoms, Figure 4 presents examples of signals $v_{0}\left(t\right)$ received at the point $x_{0}$ on various phantoms successively subjected to the three pulses shown in Figure 3 (three signals were collected in each subplot for illustrative purposes; in the experiments, only one of these pulses was used in each measurement). Waveform profiles $v_{i}\left(t\right)$ collected from all receiving points $x_{i}$ show the propagation of fast and slow wave packets over the phantom’s surface (Figure 5). The frequency spectra $v_{i}\left(f\right)=F_{t}\left[v_{i}\left(t\right)\right]$ and time–frequency wavelet images $w_{i}\left(t,f\right)=W\left[v_{i}\left(\tau \right)\right]$ of the received signals are analyzed; $F_{t}$ and W are the Fourier and wavelet transform operators in the time domain:(1)$$F_{t}\left[v\right]=\int_{0}^{\infty }v\left(t\right)e^{i\omega t}dt,\;W\left[v\right]=\int_{0}^{\infty }v\left(\tau \right)\Psi \left(t-\tau \right)e^{i\omega \tau }d\tau ,$$
$$\Psi \left(t\right)=e^{-{\left(at\right)}^{2}},\;a=2/\Delta t,\;\omega =2\pi f.$$

Differences in wave patterns in Figure 4 and Figure 5 indicate the presence of changes in the sample structures. However, it is not easy to interpret their meanings based on the unprocessed measurement data. To identify diagnostic features, the expansions of the registered surface waves in terms of GW modes should be obtained and analyzed first.

## 3. Guided Waves in Bone Phantoms

In the computer simulation, we use *M*-layered models of bone structure. The soft tissue is modeled by a homogeneous layer, while the bone itself is divided into sublayers, and a layer of bone marrow can also be added (Figure 2). The frequency spectrum u(x,f) of the displacement wave field u(x,t) generated in the phantom by a surface load q is simulated by the solution to the corresponding boundary value problem (BVP) for a forced steady-state time harmonic oscillation $ue^{-i\omega t}$ of the elastic layered structure considered (Figure 2); ω=2πf is the angular frequency, *f* is the frequency. Since the measurements are performed along the symmetry axis, we consider 2D BVPs for the in-plane displacement $u=(u_{x},u_{z})$; x=(x,z). In this case, the general representation of its solution based on the Green’s matrix [28,29] takes the following form: (2)$$u\left(x\right)=\frac{1}{2\pi }\int_{\Gamma }K\left(\alpha ,z\right)Q\left(\alpha \right)e^{-i\alpha x}d\alpha ,$$
where $K=F_{x}\left[k\left(x\right)\right],Q=F_{x}\left[q\left(x\right)\right]$, and $U=F_{x}\left[u\right]=KQ$ are Fourier symbols in the wavenumber–frequency domain (α,f); $F_{x}$ is the Fourier transform operator with respect to horizontal coordinate *x*; $k\left(x\right)=\left(k_{1}\vdots k_{2}\right)$ is the 2 by 2 Green’s matrix, and q=(0,q) is a normal surface load. Columns $k_{j}$ are the solution vectors corresponding to the surface point loads $q=\delta \left(x\right)i_{j}$, j=1,2, applied along the basic coordinate vectors $i_{1}=\left(1,0\right)$ and $i_{2}=\left(0,1\right)$; δ is Dirac’s delta function. The integration path Γ goes in the complex plane α along the real axis rounding the real poles $\zeta_{n}$ of the matrix *K* elements in accordance with the principle of limiting absorption.

Note that with a point load q=δ(x)p(t), the Fourier symbol *Q* is reduced to the frequency spectrum P(f) of the driving pulse p(t): $Q\left(\alpha ,f\right)=F_{xt}\left[q\left(x,t\right)\right]=P\left(f\right)$. And in accordance with Equation (Equation 2), only element $K_{22}$ of matrix *K* and pulse spectrum P(f) control the Fourier symbol $V\left(\alpha ,f\right)=F_{xt}\left[v\left(x,t\right)\right]$ of the signal’s field: (3)$$V\left(\alpha ,f\right)=-i\omega U_{z}\left(\alpha ,f\right)=-i\omega K_{22}\left(\alpha ,f\right)P\left(f\right).$$

The poles $\pm \zeta_{n}$ (Reζn, Imζn≥0) are zeros of the $K_{22}$ denominator, arranged in ascending order of imaginary parts: (Imζn+1≥Imζn). They are the roots of the characteristic equation
(4)$$K_{22}^{-1}\left(\alpha ,f\right)=0$$
that gives the same dispersion curves $\alpha =\zeta_{n}\left(f\right)$ as those obtained from the dispersion equation derived in the framework of the conventional modal analysis.

The residue technique reduces integral (Equation 2) to a sum of guided waves: (5)u(x)=∑n=1Nan(z)eiζnx+O(e−ImζN+1x),x>0,
$$a_{n}{=-i\mathrm{res}K|}_{\alpha =-\zeta_{n}}Q\left(-\zeta_{n}\right),\;\zeta_{0}x>>1.$$

Here, *N* is the number of terms (GW modes) retained in the expansion; it includes the contribution of all real poles $\zeta_{n}$ and, possibly, several complex ones close to the real axis; $\zeta_{0}$ is a characteristic wavenumber.

The terms of expansion (Equation 5) are source-generated GWs, where the poles denote the wavenumbers. Accordingly, cn=ω/Reζn and vn=dω/dReζn are the phase and group velocities of the corresponding GWs, and $s_{n}=1/c_{n}$ is their slownesses. The amplitude factors $a_{n}$ determine the wave energy amount carried by each guided wave. The shapes of their dependencies on depth *z* are the same as those of the modal eigenforms; however, unlike the latter, they are uniquely determined, while the eigensolutions are determined by constant factors.

Typical dispersion curve patterns for the phantoms under study are shown in Figure 6 and Figure 7. Figure 6 depicts dispersion curves in the frequency–slowness plane (f,s), which is more convenient than traditional phase velocity curves from infinity; solid and dashed horizontal lines indicate the slowness of the *P* and *S* body waves in each sublayer. To illustrate the effects of progressive bone weakening, the numerical examples are for the intact, half-drilled, and drilled-through plates ($h_{pore}=0,h/2$, and *h*). The uncoated samples ($h_{soft}=0$) are marked I, II, and III, and the same plates coated with the $h_{soft}=2$ mm soft layer are labeled IV, V, and VI, respectively; by default, h=3 mm, and other *h* cases are additionally marked.

Within the isotropic model, each sublayer is defined by its body wave velocities $c_{p}=\sqrt{C_{11}/\rho }$ and $c_{s}=\sqrt{C_{44}/\rho }$, and its density ρ. $C_{ij}$ represent the elastic moduli. The input material parameters for these samples are shown in Table 1 above.

Body wave velocities in the drilled part and its density were determined by measurements. Poisson’s ratio ν is shown here as additional information.

The drilled material is transversely isotropic with the horizontal plane of isotropy (x,y), and the velocities $c_{p}$ and $c_{s}$ in the table are for the body wave propagation in the horizontal direction. However, the numerical analysis showed a weak effect of accounting for such anisotropy; this was true even when considering factors like material viscosity or detailed layering up to multilayered sandwich-like models with internal bone marrow and external soft tissue, e.g., [47,48]. The wave patterns on the surface primarily depend on the elastic properties of the upper sublayers.

As for the influence of artificial porosity, one can see a noticeable change in the level at which the slowness curves of the fundamental modes $A_{0}$ and $S_{0}$ progress with increasing frequency in samples II and III compared to intact plate I (Figure 6, top). The same trend is observed for the higher modes, and their outlet points (cutoff frequencies) shift to the left. Thus, with the weakening, the GWs become slower, in general, which is also noticeable in the group velocity curves (Figure 7), although to a lesser extent. The soft covering results in the emergence of many additional slower GW modes with velocities (slownesses) that are practically independent of the hard sublayer porosity (Figure 6, bottom). At the same time, the slowness curves in the lower parts of these subplots keep the tendency mentioned above. Obviously, these faster modes are associated with the inner hard sublayers.

In the bone QUS, the search for diagnostic signs is conventionally focused on changes in the dispersion properties of propagating surface waves. The above-mentioned variations of theoretical dispersion curves provide some hope for their use in diagnostics. However, in practice, these curves are to be extracted from the measurement data, which is not easy in view of a strong interference from other modes whose characteristics are almost independent of the cortical bone properties. In fact, the situation is even worse because of different types of modal excitability. While the theoretical dispersion curves are independent of the source and, therefore, look equally clear on the plots, the amplitudes of source-generated ‘useful’ modes are much smaller on the surface than on the noise waves.

A theoretical excitability of GW modes by a normal point source can be estimated from the magnitude images of the $-i\omega K_{22}$ element controlling the received signals (Equation (Equation 3), P=1). In Figure 8, they are shown in the same frequency-slowness plane (f,s) as in Figure 6; s=α/ω. In these images, the dark bands follow the slowness dispersion curves depicted in Figure 6, with their width being proportional to the GW amplitude. It can be seen that, even theoretically, the curves in the bottom images of Figure 8 are poorly visible in the slowness range from about 0.5 to 1 (s/km), while here, they are the primary interests.

The dominant contribution of the new modes—emerging in the coated samples to the wave field on the surface—is also explained by their eigenforms featured by much higher oscillation amplitudes in the upper soft coating compared to the underlying harder substrate. Examples of such depth dependencies of the GW amplitude factors $a_{n}\left(z\right)$ are shown in Figure 9 (in fact, there are plots of Iman while Rean=0; $a_{n}$ are the second components of vectors $a_{n}$ in Equation (Equation 5); note that the scale of the horizontal axes in the bottom images yields ten times larger amplitude values than in the upper images). It results in the wave energy concentration of the corresponding source-excited GWs in the upper soft coating.

## 4. *H*-Function-Based Retrieval of Experimental Dispersion Curves

The *H*-function was introduced as a discrete approximation of the two-dimensional Fourier transform operator $F_{xt}$ to visualize the GW dispersion curves [37], similar to the images of $|V|=|\omega K_{22}|$ in Figure 8 above. In the examples below, we calculate it as the truncated series of the discrete Fourier transform:$$H\left(\alpha ,f\right)=2\Delta x\Delta t|\sum_{i=0}^{N_{x}}\sum_{j=0}^{N_{t}}v_{ij}e^{i\omega t_{j}}\cos\alpha x_{i}|$$(due to symmetry, the array $v_{ij}$ was augmented by the same values at the points $-x_{i}$ to the left of the source, which yielded $e^{i\alpha x_{i}}+e^{-i\alpha x_{i}}=2\cos\alpha x_{i}$).

Obviously, its accuracy is limited by the number of receiving points $x_{i}$ and time steps $t_{j}$, and depends on the numerical integration steps Δx and Δt. Therefore, the *H*-function cannot provide the same sharp images as theoretical Fourier symbols $K_{22}\left(\alpha ,f\right)$ in Figure 8, but rather blurred spots. Moreover, its frequency range is limited by the range of the driving pulse, and in this sense, most information can be obtained using sweep signals (see |P(f)| in Figure 3). Still, the dispersion curves calculated with properly chosen input parameters pass through such spots for any driving pulses (e.g., Figure 10). The spot centers indicate the local maxima of the approximated function |V(α,f)|. A reasonable agreement of the calculated dispersion curves means that the input (effective) parameters are close to the properties of measured samples.

However, the effective parameters of more complex phantoms with drilled and coated plates, not to mention real bones, are generally unknown. Conventionally, they are obtained by minimizing an objective function *F* that specifies a discrepancy between the measured and calculated dispersion characteristics of the excited GWs (phase or group velocities, wavenumbers, wavelengths, etc. [11,15,33,53]). All of them are expressed through the roots of the GW characteristic equation, which makes it necessary to solve at each *F* minimization step. The explicit Green’s matrix-based representation (Equation 2)–(Equation 5) provides a significant computational advantage over popular mesh-based simulations, such as FEM or finite difference. But the search for the roots of dispersion equation (Equation 4) still requires hundreds and thousands of calls to the procedure of the matrix *K* calculation at each step.

To reduce such computational expenses, we implemented a new form of the objective function expressed directly through the matrix *K* elements at the reference points $\left(f_{p},\zeta_{p}\right)$ in the frequency–wavenumber plane [34]: (6)$$F\left(C,\rho ,h\right)=\sum_{p}|K_{22}^{-1}\left(\zeta_{p}^{m},f_{p}^{m}\right)|$$(a similar objective function was also independently proposed in Ref. [42]).

The pairs $\left(f_{p}^{m},\zeta_{p}^{m}\right)$ are experimentally obtained points (e.g., spot centers in Figure 10) while their calculated counterparts $\left(f_{p}^{c},\zeta_{p}^{c}\right)$ are not required at all. Since $\zeta_{p}$ are the poles of $K_{22}$ at certain frequencies $f_{p}$, the corresponding terms of sum (Equation 6) turn to zero as soon as the varied input parameters reach values yielding $K_{22}$ with such poles without calculating $\zeta_{p}^{c}$ themselves.

## 5. MPM-Based Retrieving of GW Parameters

The choice of reference pairs $\left(f_{p}^{m},\zeta_{p}^{m}\right)$ from the images of *H*-functions is a method that lacks sufficient rigor and accuracy. A more accurate set of values can be derived by processing experimental data using the matrix pencil method (MPM) [38,39,40].

Based on expansion (Equation 5), the signals’ frequency spectra $v_{i}\left(f\right)$ can be written in the following form
$$v_{i}=\sum_{n=1}^{N}b_{n}\lambda_{n}^{i},\;i=0,1,\cdots ,N_{x}-1,$$
where $\lambda_{n}=e^{i\zeta_{n}\Delta x}$ and $b_{n}=-i\omega a_{n}e^{i\zeta_{n}x_{0}}$. This representation, in terms of power $\lambda_{n}$, is possible because moving to the next point $x_{i}=x_{i-1}+\Delta x$ is equivalent to multiplying its terms by $\lambda_{n}$.

In accordance with the MPM scheme, the matrix pencil
$$V\left(\lambda \right)=V_{1}-\lambda V_{0}$$
is formed from the matrices
$$V_{0}=\left(\begin{array}{cccc} v_{0} & v_{1} & \cdots & v_{L-1}\\ v_{1} & v_{2} & \cdots & v_{L}\\ \cdots & \cdots & \cdots & \cdots \\ v_{N_{x}-L-1} & v_{N_{x}-L} & \cdots & v_{N_{x}-2} \end{array}\right)$$
and
$$V_{1}=\left(\begin{array}{cccc} v_{1} & v_{2} & \cdots & v_{L}\\ u_{2} & v_{3} & \cdots & v_{L+1}\\ \cdots & \cdots & \cdots & \cdots \\ v_{N_{x}-L} & v_{N_{x}-L+1} & \cdots & v_{N_{x}-1} \end{array}\right)$$
of size $(N_{x}-L)\times L$. Their rows are composed of $v_{i}$ values at successive *L* points $x_{i}$ with a unit shift of the starting point index in each subsequent line. The number $L:N\leq L\leq N_{x}-L$ is referred to as a pencil parameter.

Under ideal conditions without noise, $\mathrm{rank}\left(V_{0}\right)=\mathrm{rank}\left(V_{1}\right)=N$, while with $\lambda =\lambda_{n}$, the rank of the matrix pencil V(λ) decreases by one [38]. That is, $\lambda_{n},\;n=1,\cdots ,N$ are among the eigenvalues. They can be found in different ways; first of all, as eigenvalues of the matrix $B_{0}=V_{0}^{+}V_{1}$ of size L×L; this is done after the pencil is multiplied by the pseudo-inverse Moore–Penrose matrix $V_{0}^{+}$ [54], or they can be derived using the singular value decomposition method (SVD) [1,10]. Then, the complex wavenumbers $\zeta_{n}=\log\left(\lambda_{n}\right)/\left(i\Delta x\right)$ are specified in the range of |Reζn|<π/Δx.

Obviously, the experimental data are imperfect; they contain noise, reflected waves, and wave interference. Moreover, some GWs are poorly excitable, so that fewer than *N*-correct eigenvalues can actually be found, while the remaining roots are induced by noise. To filter them, we first use a double-sided MPM scheme proposed in [40]. It is based on the fact that the eigenvalues $\mu_{n}$ of the matrix pencil $\mu V_{1}-V_{0}$ must be equal to $1/\lambda_{n}$. The extra roots associated with the noise are unstable, and we discard those from $\lambda_{n}$ that do not satisfy the condition $|\left(\lambda_{n}-1/\mu_{m}\right)/\lambda_{n}|<\delta$ for all *m* with some threshold level δ. Among the processed results, there were values with negative Re $\zeta_{n}<0$. These values were associated with waves reflected from the right edge and were, thus, excluded. Figure 11 presents examples of wavenumbers $\zeta_{n}$ obtained with the double-sided MPM processing of data measured on several phantoms subjected to the sweep-driving pulse.

The points remaining in the figure trace the dispersion curves, but the extra points induced by noise are still present in abundance. Therefore, we perform additional filtering against the *H*-function: only points where the condition $H(f_{p},\zeta_{p})\;<\;\varepsilon ||H||$ with a certain threshold ε holds are retained for the goal function. Thus, we discard the points related to small amplitude modes, whose findings are unstable (Figure 12).

## 6. Diagnostic Indicators

### 6.1. Effective Material Parameters

A shift in the GW characteristics signifies the onset of osteoporosis. It reflects the change in the bone’s density and elastic properties, which is a direct consequence of the disease. Therefore, the detection of some changes in the effective material parameters could serve as a direct diagnostic indicator. As discussed above, the minimization of the goal function (Equation 6) makes it possible to determine the effective parameters of a layered waveguide from the GW characteristics extracted from surface measurements. However, as demonstrated by the examples in Figure 6, Figure 7 and Figure 8, the variations in mechanical properties of internal sublayers have little effect on the surface waves. And the first question arises: is it possible, at least theoretically, to restore their effective parameters and detect their insignificant changes from surface measurements?

To clarify this question, a series of numerical experiments was carried out for various typical phantoms under study. First, the surface signals $v_{i}\left(t\right)$ were calculated using integral and asymptotic representations (Equation 2)–(Equation 5). Then, the synthetic arrays $v_{ij}$ were processed according to the general schemes described in Section 4 and Section 5, and the points $\left(f_{p},\zeta_{p}\right)$ selected from the MPM results by *H*-filtering were substituted into objective function (Equation 6). It was minimized by the coordinate-wise descent method, varying the material parameters of each sublayer separately, and assuming the rest to be known.

We attempted this approach using $v_{ij}$ arrays calculated for each of the three driving pulses p(t) shown in Figure 3; the initial values of the variable parameters $c_{p},c_{s}$, and $h_{m}$ (sublayer thickness) were taken quite far from those used in the input data. The most encouraging results were obtained with the sweep pulse: the effective parameters were fairly well reconstructed for the upper hard layer (Figure 13, left), and the latent weakened sublayer (Figure 13, right), with both uncovered and covered phantoms. The two-cycle pulses, however, do not provide the same good accuracy. For some complex-structured samples, they allow significant deviations, especially with $f_{c}=100$ kHz, which has the narrowest frequency range.

With the experimental data, the first results were rather discouraging because of the large number of local minima appearing in the objective function, once the theoretical real $\zeta_{p}$ were replaced by the MPM-extracted complex-valued ones. An additional reason was a noticeable noise-induced fluctuation of the MPM-obtained values $\zeta_{p}\left(f\right)$ (e.g., Figure 11 and Figure 12), instead of following the theoretically smooth dispersion curves. Smoothing these data and accounting for the attenuation (viscosity) of real materials in the computer model eliminated such a multiplicity of local minima.

### 6.2. Resonance Response

As indirect indications of the developing disease, it was noticed that the GWs generally become slower and their cutoff frequencies shift to the left (Figure 6). The first feature is, however, poorly distinguishable, even theoretically, while the second is well visible due to the sufficient GW excitability near the cutoffs (Figure 8). At these frequencies, the group velocities of the corresponding modes become zero (Figure 7), indicating the absence of energy transfer from the source to infinity. This leads to the appearance of so-called thickness resonances featured by the surface oscillation independent of the horizontal coordinate *x* with a zero wavenumber $\zeta_{n}$. In the frequency domain, the resonances appear as peaks in the signal’s frequency spectrum.

Similar and more powerful resonance peaks also arise at zero-group-velocity (ZGV) frequencies with non-zero wavenumbers [55,56]. The ZGV mode occurs at the lower limit of the backward-wave range [57], appearing near some cutoffs due to specific dispersion curves bending with a negative slope, resulting in a negative group velocity (loops below the abscissa in Figure 7).

In the GW expansion (Equation 5), all of these resonance frequencies $f_{r}$ are singularity points of the amplitude factors $a_{n}\left(f\right)=\left|a_{z,n}\left(0,f\right)\right|$, which manifest themselves as peaks on their plots (Figure 14).

Both resonances can be reliably detected even from single-point measurements, especially using modern laser Doppler vibrometry [55,58]. This paves the way for the development of laser-based technologies for rapidly evaluating the thickness and material constants of elastic plates [56,59]. Currently, these technologies are well developed for homogeneous plates, while the frequency responses of layered waveguides are featured by multiple ZGV resonances (e.g., see Ref. [60] and the review therein).

As the disease progresses, the resonance peaks also shift to the left following the cutoff and ZGV frequencies $f_{r}$, providing the possibility of using them as diagnostic indicators. The advantage of the resonance response method is its enhanced ability to detect these peaks compared to the dispersion curve points $\left(f_{p},\zeta_{p}\right)$. It does not require measurements at a set of points $x_{i}$ or with extensive processing of the arrays $v_{ij}$; a single signal v(t) received at a surface point can be enough for detecting resonance peaks.

In the time domain, the resonance frequencies yield long-duration oscillations, which appear as long ‘tails’ after powerful first arrivals (e.g., Figure 4). In the time–frequency wavelet images, these tails are also visible as pale horizontal stripes (plumes) at the corresponding frequencies (Figure 15). In line with the peak shift in Figure 14, these stripes shift downward as the weaker sublayer increases. This trend is observed for the uncoated samples, especially for the resonance plume at f≈0.4 MHz, and keeps for the coated ones as well.

To avoid the interference of first arrivals, Fourier and wavelet transforms (Equation 1) should be applied only to such tails. In the numerical examples, the tails were taken, starting from the time shown in Figure 15 by vertical dashed lines, providing much clearer peak patterns (Figure 16). To assess the effects of porosity and soft coating on the resonance frequencies $f_{r}$, the charts in Figure 17 visualize the changes in their values for different thickness samples, h=3 mm (top) and h=6 mm (bottom). In samples IV – VI (right subplots), the soft coating yields additional resonances at lower frequencies, weakly dependent on the properties of the bottom sublayer, while the downtrend of the other resonance frequencies is clearly visible.

## 7. Concluding Remarks

1. Computer simulations of forced guided wave propagation, using the semi-analytical Green’s matrix-based model, show that despite a noticeable variation in the theoretical dispersion curves, their use as diagnostic indicators is not very promising because of the weak excitability of the most interesting modes, reflecting changes in the internal diseased sublayers.

2. Changes in elastic moduli directly indicate the development of osteoporosis. Their monitoring is possible by solving the inverse problem of restoring effective moduli using GW characteristics extracted from the data arrays of surface measurements based on their two-sided MPM processing, *H*-filtering, smoothing, and accounting for complex-valued wavenumbers.

3. The sensitivity and consistent correlation of resonance response frequencies with the degradation of elastic properties make them a promising diagnostic indicator. Their determination is more reliable and requires much less measurement and computational costs.

## Figures and Tables

**Figure 1 materials-16-06179-f001:**
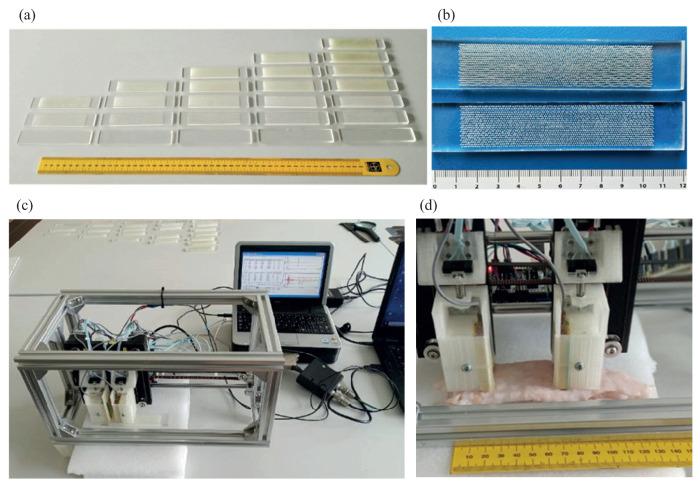
(**a**) Phantom blanks: plexiglass plates drilled from below for different depths; (**b**) top view of plates drilled in a checkerboard pattern; (**c**) experimental setup; and (**d**) setup with a specimen covered by mammalian tissue.

**Figure 2 materials-16-06179-f002:**
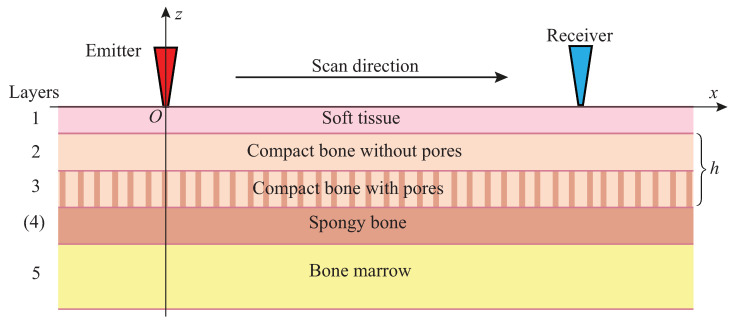
Multilayer model of the bone composite structure and measurement scheme.

**Figure 3 materials-16-06179-f003:**
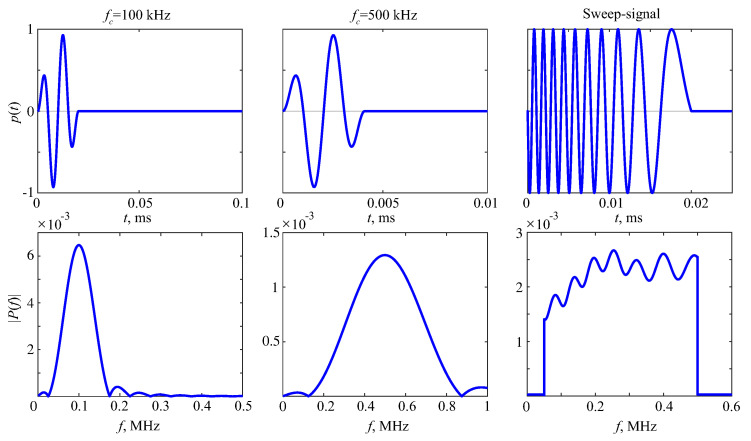
Examples of driving pulses p(t) (**top**) and their frequency spectra |P(f)| (**bottom**).

**Figure 4 materials-16-06179-f004:**
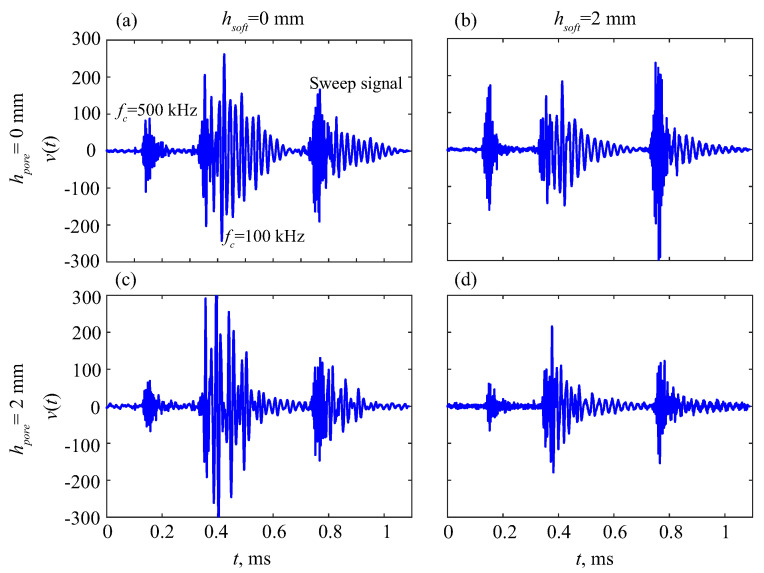
Examples of measured signals on the phantoms successively subjected to the three pulses shown in Figure 3: uncoated (**a**) and coated (**b**) samples with intact plates; and uncovered (**c**) and covered (**d**) 2/3 drilled plates; h=3 mm.

**Figure 5 materials-16-06179-f005:**
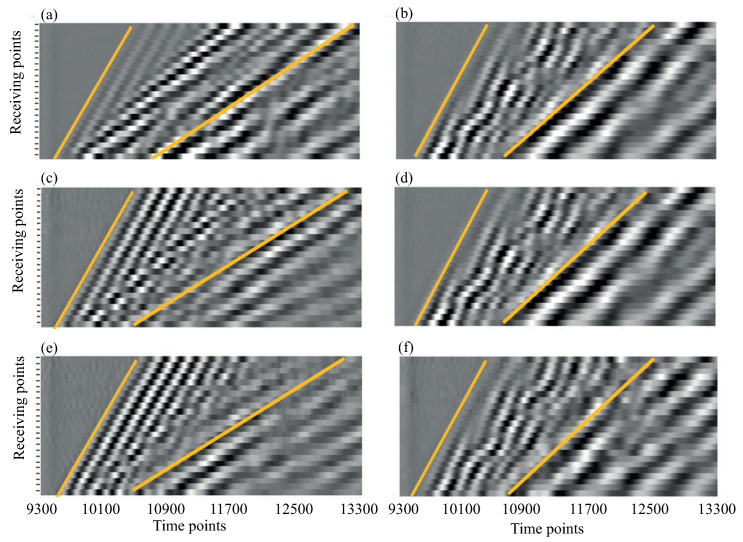
Examples of time–space waveform profiles measured at the 2/3 drilled phantoms (h= 3 mm, $h_{pore}=$ 2 mm; left column) and intact thick-plate phantoms (h= 6 mm, $h_{pore}=$ 0 mm; right column); uncoated (**a**,**b**) and coated with $h_{soft}=$ 2 mm (**c**,**d**) and $h_{soft}=$ 4 mm (**e**,**f**) soft layer; points’ spacing Δx=1 mm, time discrete Δt= 0.03 μs. Straight lines emphasize the propagation of fast and slow wave packets.

**Figure 6 materials-16-06179-f006:**
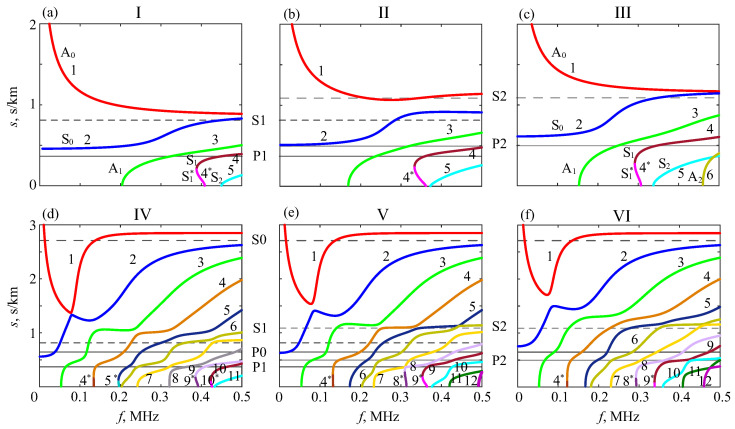
Slowness dispersion curves for uncoated samples I–III (top, (**a**–**c**)) and coated phantoms IV–VI (bottom, (**d**–**f**)); h=3 mm.

**Figure 7 materials-16-06179-f007:**
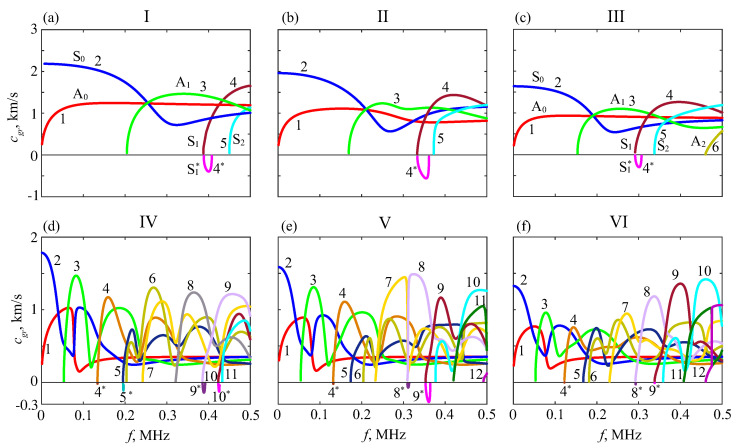
Group velocities $v_{n}$ for the same samples as in Figure 6.

**Figure 8 materials-16-06179-f008:**
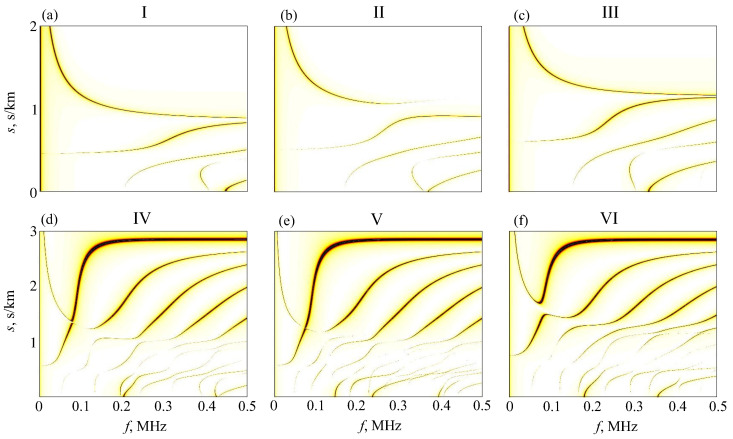
Scalogram images of the Green’s matrix element $|\omega K_{22}|$ in the frequency-slowness plane for the same phantoms I–III (**top**, (**a**–**c**)) and IV–VI (**bottom**, (**d**–**f**)) as in Figure 6 and Figure 7 above.

**Figure 9 materials-16-06179-f009:**
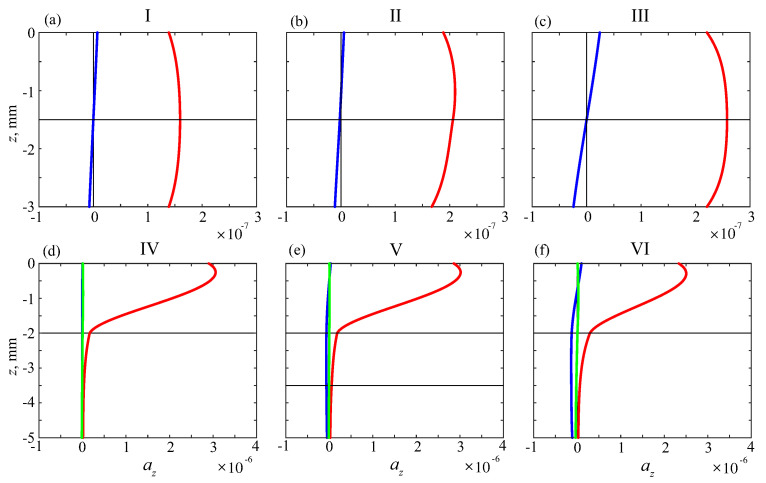
Depth dependencies of the $A_{0}$ and $S_{0}$ fundamental modes excited in uncoated plates I–III (blue and red lines in top subplots (**a**–**c**)), and of the first three modes in coated phantoms IV–VI (bottom, (**d**–**f**)), green lines are for the additional mode arising in the coated samples, horizontal black lines show interfaces between sublayers); f=100 kHz.

**Figure 10 materials-16-06179-f010:**
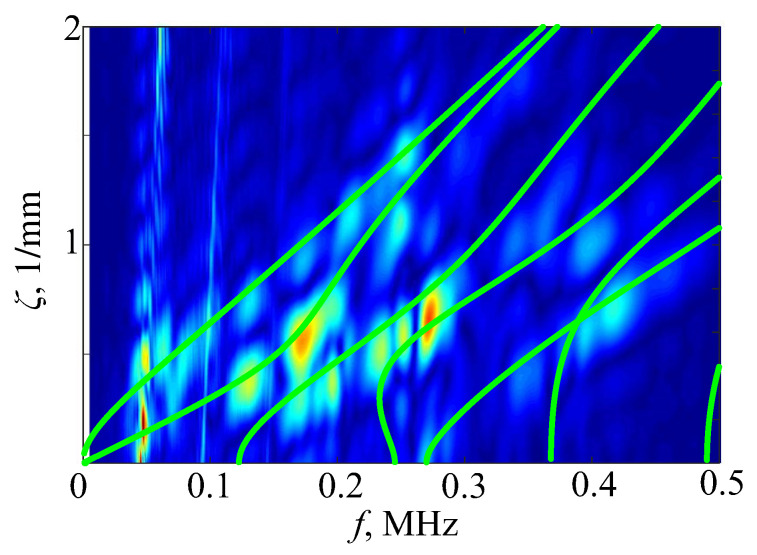
Lamb wave dispersion curves superimposed on the blurry spots of the *H*-function were calculated based on experimental data measured on the plexiglass plate of a thickness of 5 mm subjected to two pulses at $f_{c}=$ 100 and 300 kHz.

**Figure 11 materials-16-06179-f011:**
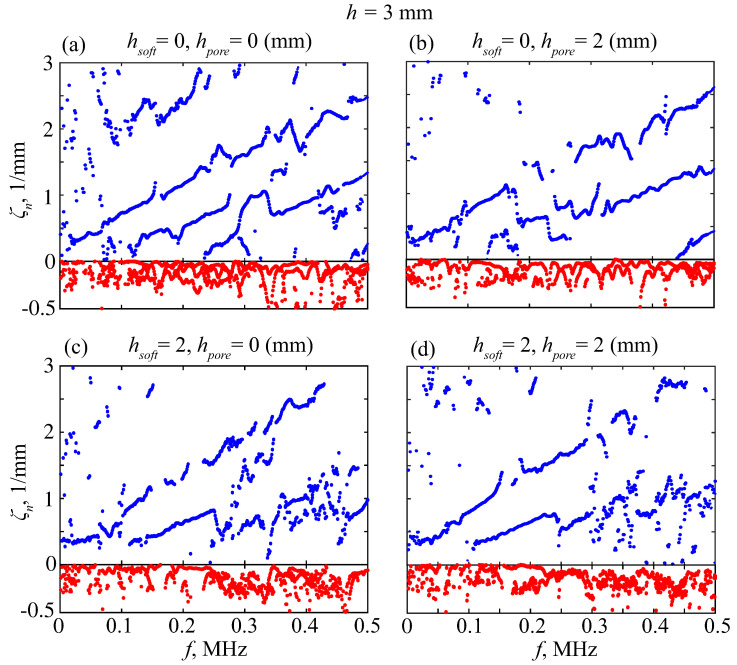
Wavenumbers $\zeta_{n}\left(f\right)$ extracted from experimental data by the double-sided MPM processing with δ=0.1; blue and red points are for Re $\zeta_{n}$ and −Im $\zeta_{n}$, respectively.

**Figure 12 materials-16-06179-f012:**
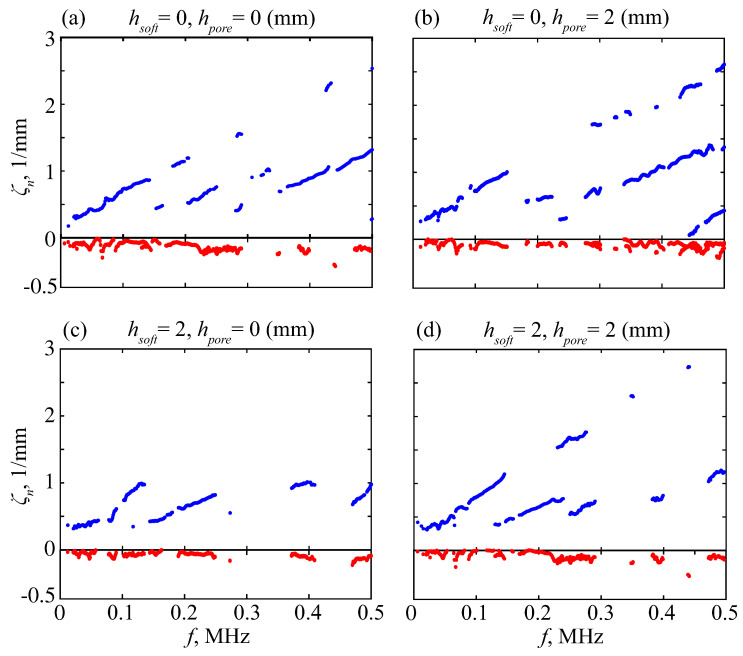
The points from Figure 11 retained after *H*-filtering with ε=0.1; blue and red points are for Re $\zeta_{n}$ and −Im $\zeta_{n}$, respectively.

**Figure 13 materials-16-06179-f013:**
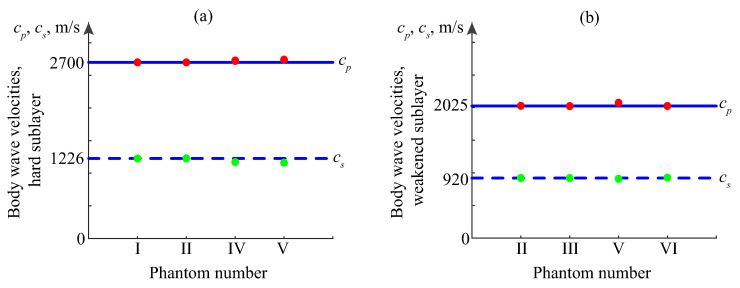
Restoring the effective material parameters of the hard (plexiglass) layer (**a**) and its lower drilled part (**b**) from synthetic data calculated for phantoms I–VI; horizontal lines indicate that the input body wave velocities $c_{p}$ and $c_{s}$ and markers are for their restored values; sweep driving pulse, Figure 3, right.

**Figure 14 materials-16-06179-f014:**
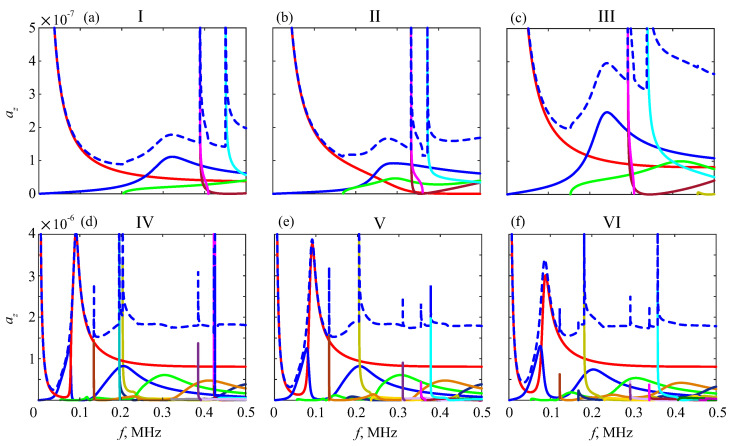
Amplitudes of frequency spectra $a_{n}\left(f\right)$ of the guided waves generated in phantoms I–VI (solid lines) and their total sum (dashed lines); delta pulse, P=1.

**Figure 15 materials-16-06179-f015:**
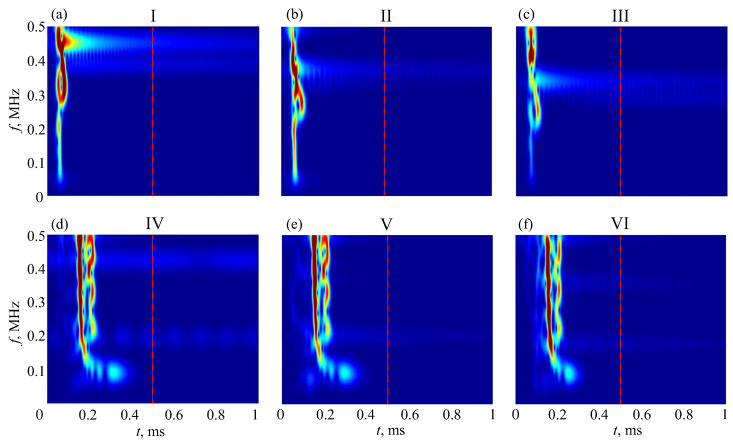
Time--frequency wavelet images $|w_{0}\left(t,f\right)|$ of the synthetic signals $v_{0}\left(t\right)$ calculated for phantoms I–VI (sweep pulse); the vertical dashed lines indicate the beginning of tails in the calculations for Figure 16 and Figure 17 below.

**Figure 16 materials-16-06179-f016:**
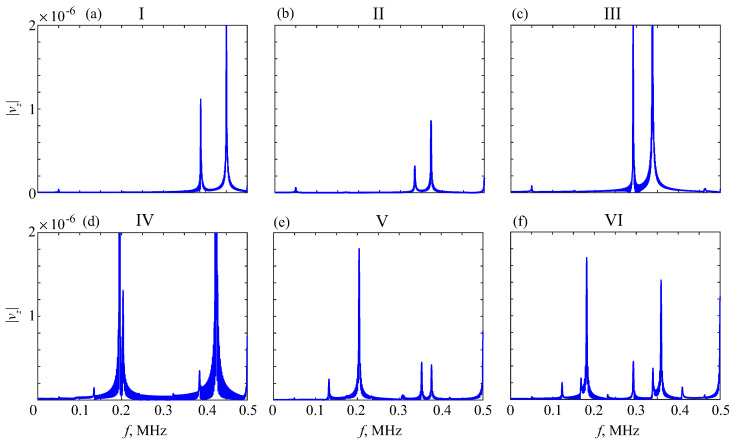
Frequency spectra of the signals’ tails for phantoms I–VI; h=3 mm.

**Figure 17 materials-16-06179-f017:**
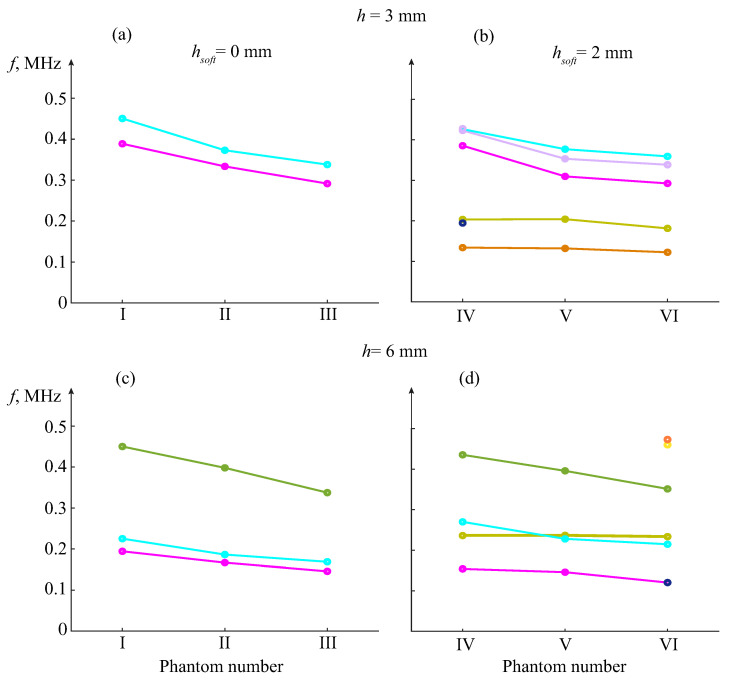
Diagrams of resonance frequencies $f_{r}$ depending on the sample structure for the peak patterns in Figure 16, uncovered (**a**) and covered (**b**) samples; the same for the *h* = 6 mm thick plate samples (**c**,**d**). Light blue and pink lines are for the resonance frequencies inherent in uncoated phantoms of thickness h=3 mm (**a**), which also appear and keep decreasing in coated samples (**b**); twice-thicker phantoms yield additional resonance (green line, (**c**)) that also keeps decreasing in covered samples (**d**).

**Table 1 materials-16-06179-t001:** Effective material parameters used in the numerical simulation.

Material	$c_{p}$ (m/s)	$c_{s}$ (m/s)	ρ (kg/m^3^)	ν
Soft plastic	1550	369	1060	0.47
Plexiglass (PMMA)	2700	1226	1190	0.37
Its drilled part	2025	920	952	0.37

## Data Availability

Not applicable.

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
