# Peer review of "Study of Ultrasonic Guided Wave Propagation in Bone Composite Structures for Revealing Osteoporosis Diagnostic Indicators"

_materials, 2023, doi:10.3390/ma16186179_

Round 1

Reviewer 1 Report

Aiming at the difficulty of signal extraction in detecting osteoporosis with the ultrasonic guided wave, this paper studies the improved matrix pencil method and the objective function based on Green matrix elements to detect the change of elastic modulus, which is used as the diagnostic index of osteoporosis. It is a very meaningful and valuable work.

Here are some suggestions for your paper:

1. The paper uses some plexiglass models to simulate osteoporosis, the reliability of the simulation experiment data, and whether the relevant conclusions support the real application scenario, I hope the author can add clarification.

2. It is suggested to add subheadings to the four subgraphs in Figure 1 to facilitate understanding of the article.

3. In line 151, a receiving point is placed for each 1mm. Are all collecting points collected at the same time? If not, does the data need to be synchronized during post-processing? How is it synchronized?

4. The received waveform in Figure 4, why does it contain three kinds of waveforms at the same time? Please give an explanation.

5. The horizontal and vertical coordinates in Figure 5 are not clearly marked and have no units.

6. The top three graphs in Figures 6 and 7 are missing horizontal coordinates and units.

7. It is recommended to add a legend in Figure 17 to illustrate the meaning of each colored line.

Reviewer 2 Report

The manuscript entitled “Study of ultrasonic guided wave propagation in bone composite structures for revealing osteoporosis diagnostic indicators” discusses the use of Green’s matrix and the analysis of bone porosity symptoms. The findings of the study are very interesting and are well-presented. Although there are some issues to be addressed before further consideration for publication. I encourage the author to revise and resubmit their manuscript. With the right corrections, I believe that this manuscript can make a valuable contribution to the field of diagnosis of osteoporosis. The scientifically valuable results obtained in this study can serve as a foundation for future research. The findings hold significant potential for practical applications in the field of regenerative medicine, as they may be relevant to tissue engineering and biomaterials. As a result, these studies can be a valuable resource for further investigations and applications aimed at improving and advancing the development of living tissues.

Minors:

Please realign the text. They should be properly placed in a line. For example, line 75

Some sentences are too long and complicated to follow. Please modify these sentences for short simple ones, for example, the last paragraph on page 4.

The figures should follow a consistent numbering and font style for better clarity and visual coherence. For example, Figure 2.

Please note that third-party copyrighted materials reproduced in your paper should as a general rule be cleared for use by the rights holders or please provide the complete statement of image permissions if approved by the responsible author. For example, the reference to Figure 2 should be added.

It's essential to add numbering to all images and provide corresponding figure numbers in the captions for better clarity and organization. This will help readers easily refer to specific images in the text. Therefore, please ensure that all images are numbered, and each caption should include the corresponding figure number to provide context and understanding for the readers.

Please add an axis title to all graphs. Only numbering of axes and columns is written in some graphs, including graphs in Figure 5.

Please include references to the formulas used in the text.

Labels for the images used in the text should be added, and all colors and lines used in the figures must have labels for better reader comprehension. For example, Figure 6.

There are many abbreviations in this manuscript; it's better to include an abbreviation table to help readers understand the meanings. This will enhance readability and comprehension of the content. 

Majors:

The use of poly(methyl methacrylate) (PMMA) plates and plastic coating on the plates is considered beneficial for modeling purposes due to their compression and artificial chemical structure. Please discusse in the introduction section. Additionally, evidence supporting the use of bone modeling with this approach should be included for further validation.

The author should extend introduction section and use and cite the following articles: https://doi.org/10.1002/jmri.27765 doi.org/10.1016/j.mri.2020.04.014

The holes created on the plexiglass plate were generated with a diameter of 0.5 millimeter. Please The method of creating these holes mention.

Please add a comparison of the properties of bone and poly(methyl methacrylate).

The empirical evidence is insufficient to support the authors’ claims. For instance, the thicknesses of the mentioned plates used ranged from 2 to 6 millimeters, and the coating thickness was up to 5 millimeters. However, specific details regarding all these thicknesses have not been mentioned in the figures and text. Please add this information for better clarity and understanding. I believe that several additional figures can improve the authors’ argumentation.

In the introduction of this research work, it is essential to provide a well-structured paragraph that sets the context and outlines the research objectives.

 I encourage the authors to provide more in-depth evidence. For instance, I would like to see more quotes on the effect of ultrasonic wave interaction regarding their density and material composition, if available.

Reviewer 3 Report

Comments and suggestions for Authors can be found in the attached document.

Can be improved.

Round 2

Reviewer 2 Report

Thank you for the revised version.